# Changing social inequalities in smoking, obesity and cause-specific mortality: Cross-national comparisons using compass typology

Andrea Teng[1]\*, Tony Blakely[2,3]\*, June Atkinson[1], Ramunė Kalėdienė[4], Mall Leinsalu[5,6], Pekka T. Martikainen[7], Jitka Rychtaříková[8], Johan P. Mackenbach[9]

1 Department of Public Health, University of Otago, Dunedin, New Zealand, 2 Melbourne School of Population and Global Health, University of Melbourne, Melbourne, Victoria, Australia, 3 Department of Public Health, University of Otago Wellington, Wellington, New Zealand, 4 Faculty of Public Health, Lithuanian University of Health Sciences, Kaunas, Lithuania, 5 Stockholm Centre for Health and Social Change, Södertörn University, Huddinge, Sweden, 6 Department of Epidemiology and Biostatistics, the National Institute for Health Development, Estonia, Sweden, 7 Population Research Unit, Faculty of Social Sciences, University of Helsinki, Helsinki, Finland, 8 Faculty of Science, Charles University, Prague, Czech Republic, 9 Department of Public Health, Erasmus MC, Rotterdam, Netherlands

\* andrea.teng@otago.ac.nz (AT); ablakely@unimelb.edu.au (TB)

**Data Availability Statement:** Study findings data are available in S2 and S3 supplementary files. Underlying datasets may be available on request

## Abstract

### Background

In many countries smoking rates have declined and obesity rates have increased, and social inequalities in each have varied over time. At the same time, mortality has declined in most high-income countries, but gaps by educational qualification persist—at least partially due to differential smoking and obesity distributions. This study uses a compass typology to simultaneously examine the magnitude and trends in educational inequalities across multiple countries in: a) smoking and obesity; b) smoking-related mortality and c) cause-specific mortality.

### Methods

Smoking prevalence, obesity prevalence and cause-specific mortality rates (35–79 year olds by sex) in nine European countries and New Zealand were sourced from between 1980 and 2010. We calculated relative and absolute inequalities in prevalence and mortality (relative and slope indices of inequality, respectively RII, SII) by highest educational qualification. Countries were then plotted on a compass typology which simultaneously examines trends in the population average rates or odds on the x-axis, RII on the Y-axis, and contour lines depicting SII.

### Findings

**Smoking and obesity**. Smoking prevalence in men decreased over time but relative inequalities increased. For women there were fewer declines in smoking prevalence and relative inequalities tended to increase. Obesity prevalence in men and women increased over time with a mixed picture of increasing absolute and sometimes relative inequalities.

depending on individual country requirements. Data cannot be shared publicly because of individual country restrictions. Data are available from each country for researchers who meet the criteria for access to confidential data. Contact information is in Table S5d. The authors had no special access privileges to the data others would not have.

**Funding:** AT was funded for this study by the University of Otago, Wellington (Dean's Grant). This study was supported by a grant (FP7-CP-FP grant no. 278511) from the European Commission Research and Innovation Directorate General, as part of the "Developing methodologies to reduce inequalities in the determinants of health" (DEMETRIQ) project. Funders played no role in study design, data collection and analysis, decision to publish or preparation of the manuscript.

**Competing interests:** The authors have declared that no competing interests exist.

Absolute inequalities in obesity increased for men and women in Czech Republic, France, New Zealand, Norway, for women in Austria and Lithuania, and for men in Finland.

**Cause-specific mortality**. Average rates of smoking-related mortality were generally stable or increasing for women, accompanied by increasing relative inequalities. For men, average rates were stable or decreasing, but relative inequalities increased over time. Cardiovascular disease, cancer, and external injury rates generally decreased over time, and relative inequalities increased. In Eastern European countries mortality started declining later compared to other countries, however it remained at higher levels; and absolute inequalities in mortality increased whereas they were more stable elsewhere.

## Conclusions

Tobacco control remains vital for addressing social inequalities in health by education, and focus on the least educated is required to address increasing relative inequalities. Increasing obesity in all countries and increasing absolute obesity inequalities in several countries is concerning for future potential health impacts. Obesity prevention may be increasingly important for addressing health inequalities in some settings. The compass typology was useful to compare trends in inequalities because it simultaneously tracks changes in rates/odds, and absolute and relative inequality measures.

## Introduction

In many countries the prevalence of smoking is declining and obesity is increasing, but the social patterning of these risk factors has varied over time. There are important implications for inequalities in premature mortality, given the strong link of both smoking and obesity to premature mortality.

Smoking prevalence has been declining over time in many countries but still remains a leading cause of death. These declines have tended to be greatest in men and women who have a tertiary education [1–3], and future widening of social inequalities in smoking-related disease has been predicted [1]. In many settings, tobacco smoking is the major contributor to social inequalities in premature mortality [4, 5]. For example the Preston-Glei-Wilmoth method relies on lung cancer mortality as an indicator of the accumulated population exposure to smoking and was used to estimate the contribution of smoking to absolute educational inequalities in all-cause mortality in 14 European countries from 1990–2004 [5]. Between countries the contribution ranged from 19% to 55% in men, and −1% to 56% in women [5]. In Norway the contribution of smoking declined over time in men but increased in women [6]. In New Zealand smoking has been estimated to contribute to 21% and 11% of relative social inequalities in mortality in men and women respectively by education [7] with changes over time [8]. Tobacco control is considered one of the key entry points for addressing educational inequalities in mortality.

Recent decades have also seen ongoing increases in obesity prevalence in many countries. Obesity is significantly associated with all-cause mortality [9] for example by increased rates of diabetes, cardiovascular disease and cancer. Where there is strong social patterning of obesity it is likely to make an increasing contribution to health inequalities [10]. A European study used population attributable fractions and rate ratios from the literature to estimate how a

scenario elimination of differences in obesity by education would decrease relative inequality in all-cause mortality. This resulted in average declines of 6% in men and 16% in women in relative inequalities [11]. There was variation between countries from 0% to 12% in men and 4% to 42% in women (eg lower in some Eastern European countries and higher in some Southern European countries) [11]. These figures were greater for causes of death such as diabetes and ischaemic heart disease. In another study, obesity was a less important contributor to income inequalities in health status in the UK because obesity was more equally distributed by income [12].

The world has witnessed dramatic declines in mortality in recent decades (albeit with slowing in some groups recently), however relative inequalities in premature mortality by socioeconomic position have continued to increase in many European countries [13], New Zealand [14] and elsewhere. There is variation between countries in the magnitude of these socioeconomic inequalities. Relative inequalities in mortality by education have continued to rise in most European countries but not in Southern Europe, where inequalities in mortality are smaller [13]. Eastern Europe countries have experienced larger increases in relative inequalities over time than other European countries [13]. At least part of the differences in these effects of education on mortality is likely to be due to differences in the prevalence rates of smoking and obesity [15].

This study's authors have previously published comparisons of education inequalities in cause-specific premature mortality in Europe (1990–2010) [13, 16], and New Zealand (1981–2011) [14]. Given the potential confusion and challenge of understanding trends over time in average rates and both absolute and relative inequalities, the later study developed a compass typology. Specifically, graphs of average rates on the x axis and relative inequalities on the y axis are used. For any given point on this x-y plane, absolute inequalities are given by contour lines, according to the mathematical interdependence of these quantities. Country trends are plotted over time, applying the compass typology. The most desirable direction of travel is southwest, because it represents decreasing average rates and decreasing relative and absolute inequalities. The most undesirable direction is northeast, with all three quantities increasing over time. These graphs efficiently convey a lot of information without requiring several separate graphs of mortality rates and inequality trends. Empirically for all-cause mortality, the common pattern across countries is of decreasing mortality rates over time, with increasing relative inequalities but stable absolute inequalities: a northwest direction of travel [14]. In this current paper, we apply this compass typology for the first time to risk factors (smoking and obesity) and country-level comparisons of cause specific mortality rates.

Cross-country comparisons of trends provide feedback on comparative progress, insight into the potential malleability of these trends over time and they may stimulate further hypotheses about national level political, economic and social drivers of health and health inequalities. The compass typology can also predict the expected future direction of travel for both mortality and risk factors, given current settings and compass trajectories. For example inequality trends in smoking and obesity can help predict the likely future trends in smoking- and obesity- related mortality.

## Aims and objectives

This study aimed to describe the simultaneous overall trends and trends in educational inequalities for smoking prevalence, obesity prevalence and cause-specific mortality, comparing adults in nine European countries and New Zealand using compass typology.

**Table 1. Country data for socioeconomic inequalities in mortality and smoking/obesity prevalence by year.**

| Region | Country | Data | Years covered by the analysis | | | | | | |
|---|---|---|---|---|---|---|---|---|---|
| | | | 1980s | | 1990s | | 2000s | | 2010s |
| Australasia | New Zealand | Mortality | 1981–84 | 1986–89 | 1991–94 | 1996–99 | 2001–06 | 2006–11 | |
| | | Smoking/obesity | | | | 1996-97[a] | 2002–03 | 2006–07 | 2011–12 |
| Northern Europe | Finland | Mortality | 1981–85 | 1986–90 | 1991–95 | 1996–00 | 2001–05 | 2006–10 | |
| | | Smoking/obesity | | | 1993/95 | 1997/99 | 2001/03 | 2005/07 | 2009/11 |
| | Norway | Mortality | 1980–85 | 1985–90 | 1990–95 | 1995–01 | 2001–06 | 2006–09 | |
| | | Smoking/obesity | | | | 1998 | 2002 | 2005, 2008 | |
| Western Europe | Austria | Mortality | 1981–82 | | 1991–92 | | 2001–02 | | |
| | | Smoking/obesity | 1983[b] | | 1991 | 1999 | | 2006 | |
| | France | Mortality | 1980–81, 1982–86 | 1987–89 | 1990–94 | 1995–98 | 1999–03 | 2004–07 | |
| | | Smoking/obesity | 1980[b] | | 1991 | | 2000 | 2005 | 2010 |
| | England and Wales | Mortality | 1976–81 | | | | 2001–06 | 2006–09 | |
| | England | Smoking/obesity | 1980 [ab] | 1986 [ab] | 1990 [a] | 1996 [a] | 2000 [a] | 2005 [a] | 2010 [a] |
| Eastern Europe | Czech Republic | Mortality | 1982–85 | | | | 1998–03 | | |
| | | Smoking/obesity | | | 1993/99 | | 2002/08 | | |
| | Estonia | Mortality | | 1987–91 | | | 1998–02 | | |
| | | Smoking/obesity | | | | 1996 | | 2006 | |
| | Hungary | Mortality | 1978–81 | 1988–91 | | | 1999–02 | | |
| | | Smoking/obesity | | | 1994 | | 2000, 2003 | 2009 | |
| | Lithuania | Mortality | | 1988–90 | | | 2001–05 | 2006–09 | |
| | | Smoking/obesity | | | | 1994/2000 | | 2006/10 | |

1. [a]Smoking data was available but not obesity data. [b]This data was available but not presented in the typology plots because it did not align with time periods of data from other centres. '/' indicates a combination of two different datasets from different years, and in this case the midpoint of the years was used to allocate the dataset to the right time period.

## Methods

National trends in smoking and obesity prevalence over two decades in 1990–2010, and cause-specific mortality from 1980–2010 (with adjacent data from late 1970s or early 2010s if available), were used. Ten countries were included in this analysis based on the availability of smoking prevalence and mortality data by education. Countries were selected if, as a minimum, they had a smoking data point in both the 1990s and 2000s; and they had a mortality data point available for both the 1980s and 2000s (eg, mid date of a cohort study), ie, in the first and last decade of the study period 1980 to 2010 (Table 1). One country was missing obesity data (England) and one country had only 2000s data for obesity (New Zealand). Of the 10 countries selected, 4 were from Eastern Europe, 3 were from Western Europe, 2 were from Northern Europe, and 1 was from Australasia (New Zealand).

### Datasets

Smoking and obesity prevalence data for men and women were collected from nationally representative surveys that also had measured highest educational qualification. Smoking was measured as current tobacco smoking except for Austria and New Zealand where it was daily smoking. This was done for 30–79 year olds, except for in some countries where upper age limits were lower (France 74 years old, and England 69 years old) [17]. Obesity was measured as a BMI of $\geq 30 kg/m^2$ calculated from self-reported height and weight except for New Zealand where these values were measured. This was done for survey participants 30–79 years old,

with the exception of Hungary and Lithuania where the upper age limit was 64 years old. Standard survey weights were not available for all surveys. Therefore results from European countries present unweighted analyses and these are expected to be similar to survey weighted results [17, 18]. Survey-weighted prevalences were used for New Zealand data. Some survey results were combined to improve study precision, for example each pair of successive surveys in Finland was pooled to calculate the mean smoking and obesity prevalence.

Education was selected as a proxy for socioeconomic position given its widespread availability; ease of harmonization across countries and lower sensitivity to health-related social mobility than other socioeconomic measures. The International Standard Classification of Education, 1997 was used to define the levels of education in all countries. Categories were 0–2 (pre-primary, primary, and lower secondary education) as "low," 3–4 (upper secondary and post-secondary non-tertiary education) as "middle," and categories 5–6 (first and second stage of tertiary education) as "high". The proportion of the population with unknown education level varied between 0% and 11% across countries, and this group could not be included in this analysis.

Cardiovascular disease (CVD), cancer, external causes of death, other causes of death, and an overlapping group of tobacco-related mortality (lung cancer, chronic obstructive pulmonary disease [COPD] and laryngeal cancer, see S3 Table for ICD codes) were examined in men and women from nine European countries[19] and New Zealand (New Zealand Census-Mortality Study). Obesity related mortality was not examined because of the likely inconsistencies in diabetes as a coded cause of death over time and between countries. All mortality data covered complete national populations with the exceptions of England and Wales (1% representative sample) and France (1% sample of individuals born in and living in mainland France). The target age range for mortality was 35–79 years, with some slight differences in New Zealand (35–74 y), Norway (40–79 y), and Lithuania (35–69 y). Mortality studies were longitudinal in design with a population census linked to mortality during a follow-up period; with the exceptions of Czech Republic, Estonia, Hungary and Lithuania (in 1988–90 and 2000–02) which used a cross-sectional design where educational level of deceased individuals was taken from death certificates and person-years by education from the population census. For Lithuania, longitudinal data were used for 2001–2005, and 1988–1990 cross-sectional data were adjusted downwards to adjust for an overestimation of inequalities in mortality; details described elsewhere [19]. Laryngeal cancer was not included in tobacco-related mortality for New Zealand but is unlikely to have much impact given these cancers were equivalent to only 1.5% of the number of lung cancer deaths in 2014. For more information on the source mortality [13, 14, 16] and survey risk factor [17, 18] datasets please see previous publications and S5 Tables.

## Analysis

Mortality rates and risk factor prevalences were age-standardised to the WHO European standard population (1976). Prevalences were converted to odds so they were on the same scale as the logistic regression used for calculating inequality measures. Average odds were calculated from the age-standardised prevalence of smoking (or obesity) divided by 1 –prevalence of smoking. Inequality estimates were based on national smoking and obesity prevalence odds (prevalence / [1—prevalence]) and mortality rates (per 100,000) by level of education.

Inequalities in mortality and prevalence were assessed with the relative (RII) and absolute (SII) indices of inequality. The relative index of inequality (RII) can be interpreted as the rate ratio of mortality/ratio of prevalence odds between those with the very lowest (rank = 1) and those with the very highest educational position (rank = 0) in the population. The slope index

of inequality (SII) represents the absolute version of the RII; ie the difference in mortality rates/prevalence odds between those with the very lowest and the very highest educational position in the population. For smoking and obesity, RIIs were calculated on the odds scale using logistic regression analysis and exponentiation of the beta coefficient and its confidence intervals from the analysis (Eq 1). SIIs were calculated from the regression-predicted odds where x = 0 and x = 1.

$$Log_{(odds\ prevalence)} = \text{ß}_{(cumulative\ smoking\ midpoints\ by\ education)} + age_{(5yr\ groups)} \tag{1}$$

Compass plots have previously been used only for rates [14]. In this paper, the compass plots were extended to work with risk factors which have a prevalence bound between 0 and 1. To avoid modelling risks less than 0 or more than 1, we used logistic regression. To calculate the SII contours for the typology plots we derived the mathematical relationship between odds of smoking (or obesity), the SII of odds of average smoking in population and the RII (on the odds ratio scale) (Eq 2, see the Supplementary Material for derivation).

$$odds = \frac{SII}{exp(0.5 \times ln[RII]) - \frac{1}{exp(0.5 \times ln[RII])}} \tag{2}$$

Mortality RIIs were calculated using Poisson regression with the observed number of deaths as the dependent variable, person-years of population as the offset variable, and educational rank and dummy variables for age (in 5 year age groups) as independent variables [20]. Educational rank was calculated as the cumulative midpoints of each education category by country, time period, sex and age group. Mortality RIIs from New Zealand were calculated slightly differently using weighted (by person time) linear regression with the standardised-mortality rate as the dependent variable, and educational rank as the independent variable, calculated separately by age group, but pooled across sex and ethnic groups [21–26]. The mortality SII was calculated as follows using formula (Eq 3), where ASMR is the age-standardised mortality rate [20].

$$SII = \frac{2 \times ASMR \times (RII - 1)}{(RII + 1)} \tag{3}$$

## Compass typology plots

For the risk factor compass plots, the log of the odds of smoking or obesity was presented on the x-axis, with the log of the RII (odds ratio scale) on the y-axis, and the SII as contours. Mortality compass plots were presented with the log of the age-standardised mortality rate on the x-axis, the log of the RII (relative scale) on the y-axis, and the SII as contours. The mathematical relationship between mortality rates, RII and SII has been reported elsewhere [14].

If compass typology trends continued on the same trajectory into the future then typology results could be used to predict future inequalities. Furthermore, given the strong relationship between smoking and smoking-related mortality, patterns in smoking inequalities may predict future smoking-related mortality inequalities for example in three decades time.

## Results

### Smoking

Fig 1 compares smoking and obesity trends in the 1990s and 2000s across ten countries by sex. The average smoking odds in men declined in most countries (less so in Czech Republic),

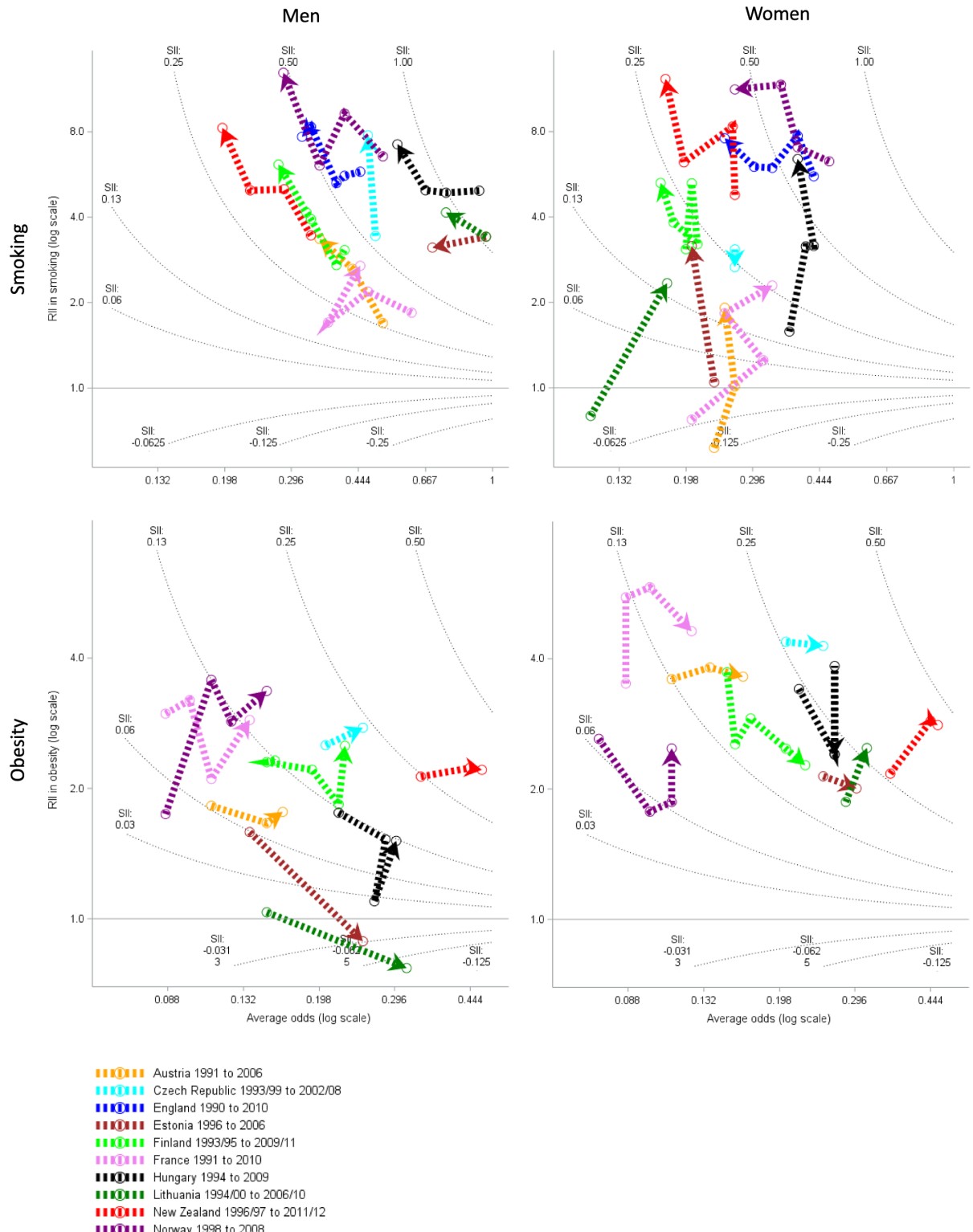

**Fig 1. Trends and education inequalities in smoking prevalence (above) and obesity prevalence (below) 1990–2010.** All countries had 1990s and 2000s data for smoking. For obesity NZ had only 2002/03 data and onwards, and England had no data, but all other countries had at least one time point from the 1990s and 2000s decade. Relative inequalities are measured by relative index of inequality (RII) on the log scale, and absolute inequalities by the slope index of inequality (SII) indicated by the contour lines.

accompanied by increased relative inequalities (RII, northwest direction of travel, except Estonia). At the end of the study period the highest smoking odds in men were in Lithuania, Estonia and Hungary and the lowest smoking odds were in New Zealand, Finland and Norway. The trend for women was generally more northward with increasing relative inequalities (except for Czech Republic), a mixed picture for absolute inequalities (which declined for England and Norway and increased in Austria France, Estonia, Hungary, and Lithuania), but generally few decreases (England, Finland, New Zealand, Norway) in smoking odds and therefore prevalence. This is possible where smoking rates in women were falling in high education groups and increasing in low educational groups. Smoking odds (and therefore prevalence) were consistently higher in men than in women. At the end of the study period the highest smoking odds in women were in Hungary, France then Norway and the lowest smoking odds were in Lithuania, Finland and New Zealand.

## Obesity

Trends for obesity prevalence differed from smoking. In every country obesity odds increased in men and women (east) with a mixed picture of increasing absolute and sometimes relative inequalities. Absolute inequalities in obesity increased for men and women in Czech Republic, France, New Zealand, Norway, and for men in Finland and for women in Austria and Lithuania. Absolute inequalities decreased for men in Lithuania and Estonia where the more educated had greater odds of obesity. There were increases in relative inequalities in men in Norway, and women in France, Lithuania and New Zealand. Obesity prevalence appeared to be similar between men and women.By the end of the study period, the highest odds of obesity were in New Zealand and Lithuania and the lowest odds of obesity were in France, Norway and Austria.

## Smoking-related mortality

Trends in smoking related mortality (eg lung cancer and COPD) were generally north/northwest in men (declining or stable [Estonia, Hungary and Norway] mortality rates and increasing relative inequalities) and north/northeast for women (stable or increasing mortality rates and increasing relative inequalities) (Fig 2). Eastern Europe countries tended to have lower relative inequalities for smoking mortality in women than other regions, crossing over from smoking mortality being greater in more educated women in the 1990s to being greater in the least educated women in the 2000s. The lowest smoking related mortality rates in men were in Norway and the highest were in Hungary. The lowest smoking related mortality rates in women were in Lithuania and the highest were in New Zealand which also experienced the highest absolute and relative inequalities. Women had lower smoking-related mortality rates than men, who also had much greater absolute and relative inequalities.

## Cause-specific mortality

Fig 3 compares the CVD, cancer, external injuries and other mortality rates and inequality trends from the 1980s to the 2000s. CVD mortality rates declined and relative inequalities increased substantially (northwest) for Western and Northern European countries and New Zealand. There was a decrease in absolute inequalities in men and women from England and Wales, Finland, and France and in women from Austria and New Zealand. Conversely countries from Eastern Europe (Hungary, Czech Republic, Estonia, Lithuania) generally followed a more northward pattern with limited declines in CVD mortality and increased absolute and relative inequalities. Mortality rates varied significantly over time and between countries ranging from 200 to 1200 per 100,000 in men and 80 and 660 in women. Mortality rates were 1.2 to

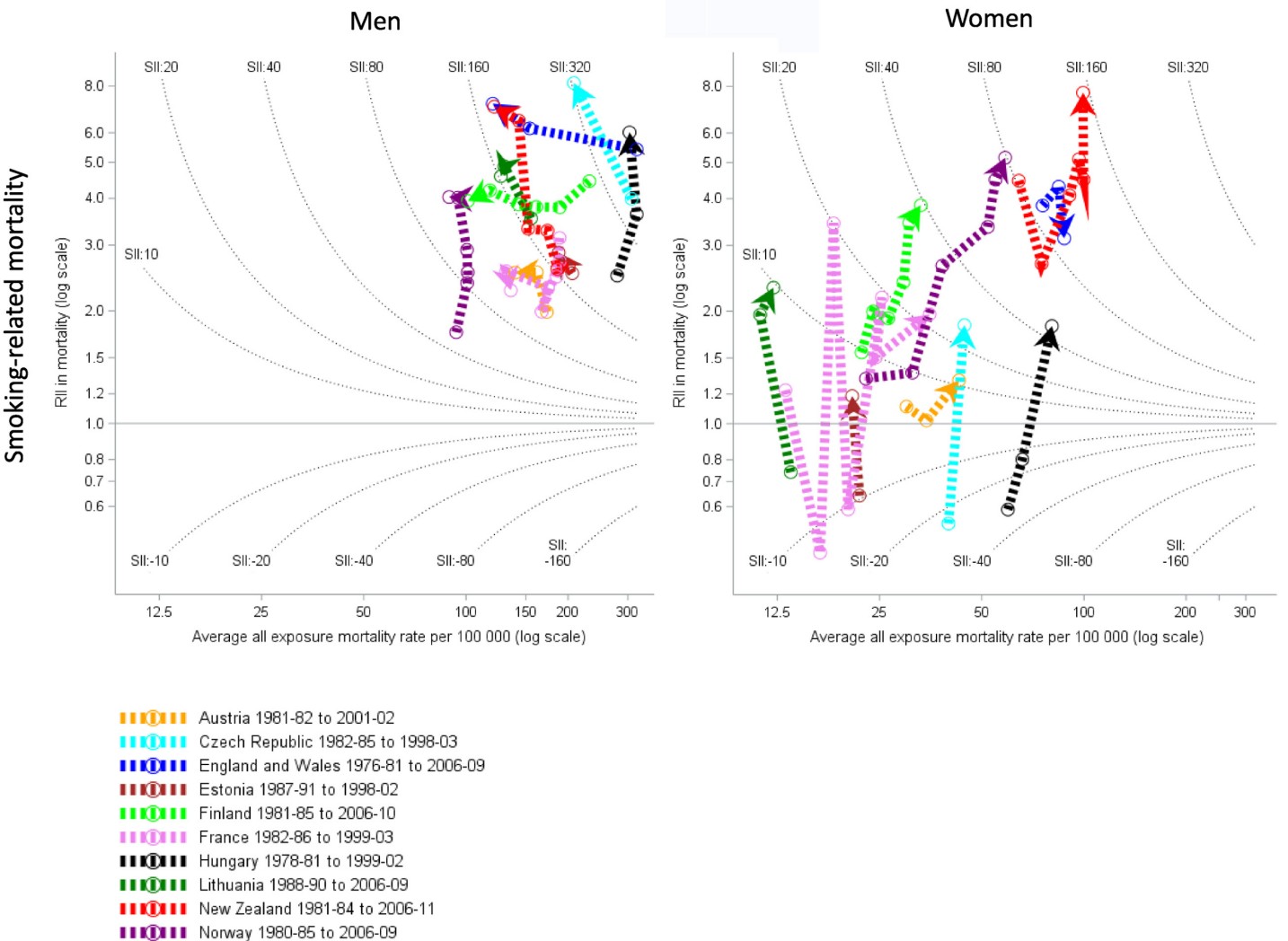

**Fig 2. Trends and education inequalities in smoking-related mortality (lung cancer, COPD and laryngeal cancer), 1980–2010.** Relative inequalities are measured by relative index of inequality (RII) on the log scale, and absolute inequalities by the slope index of inequality (SII) indicated by the contour lines.

6 times greater (RII) in the least educated, with absolute inequalities (SII) of between 80 to 1280 deaths per 100,000 population. By the end of the time period, men and women in France had the lowest CVD mortality rates and absolute inequalities.

Cancer mortality trends and regional patterns were similar to CVD with widespread declines in cancer mortality. However, men and women in Hungary and Czech Republic; men in Estonia; and women in New Zealand and Norway had generally high and stable cancer mortality and increasing absolute and relative inequalities (northward trend). There was a decrease in absolute inequalities in cancer for men and women in England and Wales, and for men in France.

External injury mortality rates were lowest in England and Wales and highest in Eastern European men and women. There was a general northward trend with stable mortality and increasing relative inequalities. However there were some declines in external injury mortality in men and women in England and Wales and Austria, men in Finland and women in France,

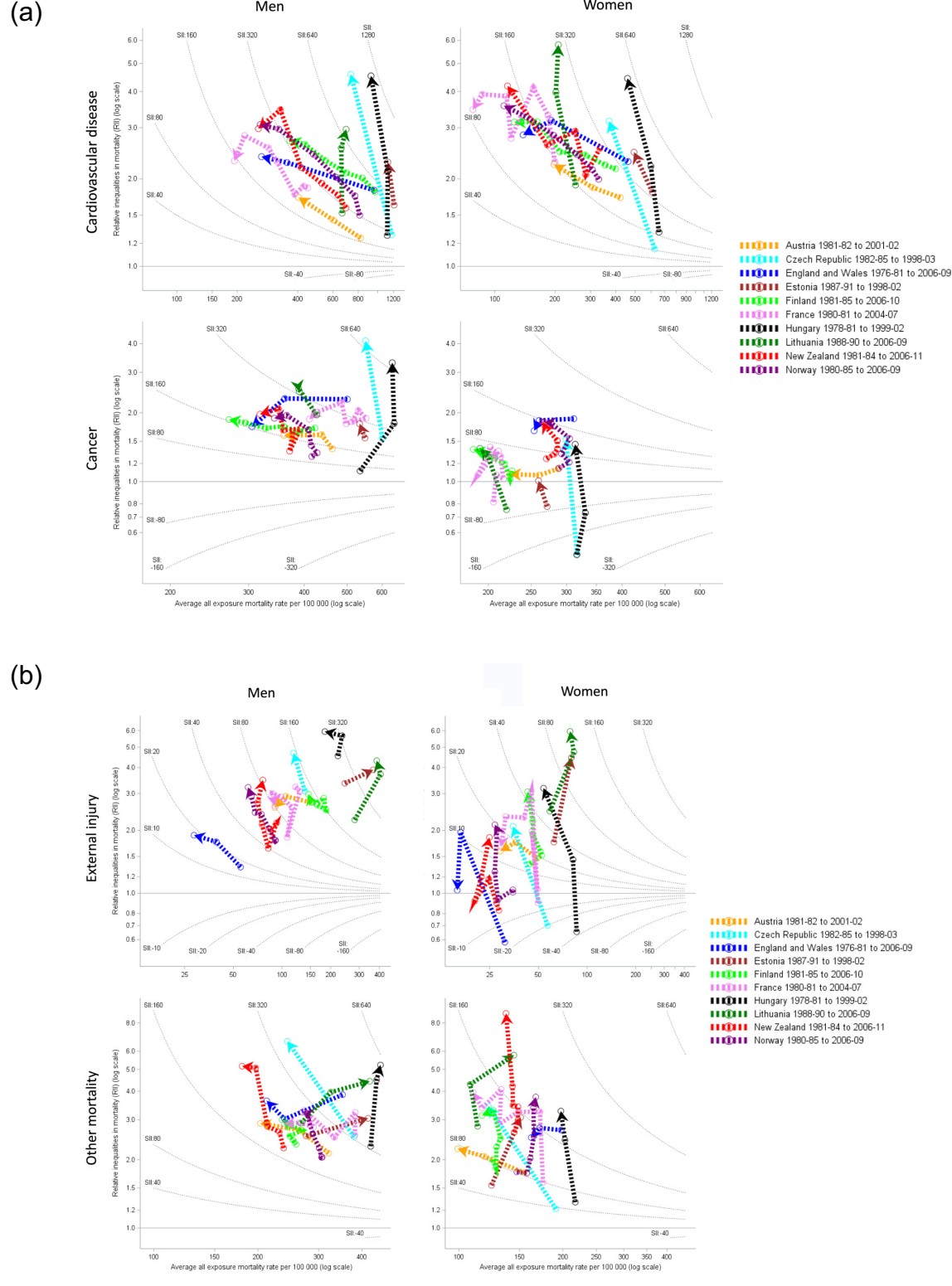

**Fig 3. Trends and education inequalities in cause-specific mortality, 1980–2010.**

Czech Republic and Hungary. External injury rates increased in Estonia and Lithuania particularly in men. All mortality rates were greater in men than in women.

## Discussion

Smoking prevalence was generally (northwest) decreasing in odds with increasing relative inequalities for men, but only a few countries had decreased odds in women (England, Finland, New Zealand and Norway). Trends in smoking-related mortality were consistent with a delay in the smoking epidemic in women and a lag compared to smoking prevalence trends. Namely among men there was a more stable smoking-related mortality (northerly) pattern compared to decreasing (northwest) smoking prevalence, and among women (northeast) there was increasing mortality compared to a generally stable (northerly) smoking prevalence. An approximately 30 year or more [27, 28] time lag from smoking to mortality (largely lung cancer and COPD deaths) has been described. Different countries appeared to be at different stages of progression the smoking epidemic, with almost all heading towards lower smoking prevalence but high relative inequalities. If existing trends continue then smoking-related mortality in men and then women will shift northwest in the decades ahead, with decreased smoking-related mortality rates and increased relative inequalities.

Obesity prevalence increased over time similarly in men and women. Indeed, no country recorded a decline in obesity prevalence from the 1990s to 2000s. This was combined with increasing absolute inequalities in many countries (eastward trend) with two countries reporting a decrease in absolute and relative inequalities where obesity became more prevalent in men with higher education (Lithuania and Estonia) but this pattern was not noted for woman. This finding has been seen previously in Estonia [29] and the authors' hypothesise physical activity from manual occupations as a contributing factor. It also may be related to being on an earlier stage in the nutrition transition [30], where energy dense food became more accessible to the most educated first after the dissolution of the Soviet Union, then later became more concentrated in the least educated. Men in Norway and women in France, Lithuania and New Zealand had a pattern of worsening absolute and relative inequalities. These trends are an increasing concern for countries and a warning about the future obesity-related disease burden and the likely increasing absolute inequalities in obesity-related mortality (such as diabetes, obesity-related cancers and cardiovascular disease). There is a concern that increasing obesity will hinder future progress in reducing obesity-related mortality and impact on inequalities in obesity-related mortality. Authors have suggested that the lag time between obesity prevalence and mortality may be decades [31–33], however there is also evidence that sudden changes in nutrition interventions can reduce CVD events within months [34]. If the greatest mortality impacts are after a long period of obesity exposure, then the trends here in obesity prevalence and absolute inequalities are not yet evident in the reported mortality rates.

Clear northwest trends emerged from the cause-specific mortality analysis, with decreasing rates of CVD, cancer, external injury, other mortality rates, while at the same time there were almost ubiquitous increased relative inequalities as described previously [13, 14, 16] but that are likely to worsen if current trends are maintained. Cause specific mortality rates in men and women from Eastern Europe countries (Hungary, Estonia, Lithuania and Czech Republic) tended more northward recording much small declines in average mortality if any and larger increases in relative inequalities than other countries.

Mortality trends reflect the social drivers of health and progress taken to reduce mortality, and how almost universally the most educated have benefited more proportionally from these improvements, thus increasing the relative gap. This is possible even when absolute mortality

gains were the same for both groups, which was the case in countries that tracked parallel to SII lines on the compass plot.

Despite the large differences in starting and finishing positions, the similarities in trajectories identified in the compass plots between countries were striking. The generally north/northwest smoking prevalence trends, the eastward obesity prevalence and the northwest mortality patterns suggest global drivers are important such as the globalisation of the obesogenic environment. The trends presented here may be similar to those found in other developed countries.

## Strengths and limitations

For the first time the compass typology is applied to country-level comparisons of smoking and obesity prevalence, cause-specific mortality, and their corresponding relative and absolute inequalities. The compass approach goes a step further than conventional ways of analyzing trends in health inequalities. For example, this paper shows the differences between countries in the relative positioning of obesity, obesity inequalities and trends, and the possible future direction of those trends, which are not clearly demonstrated using conventional methods [18].

There are however some limitations. We did not have the required data to include any countries from Southern Europe or any other regions, where trends may differ. Similarly we did not have data for the precise corresponding timespans for included studies. This was largely managed by ensuring all countries had data available in the first and the last decade of each timeframe. Also for very complex and large international comparisons, there was no possibility to collect more recent data.

Differences between data sources may have impacted some results. For example cross sectional data is used from Eastern European countries, which may have introduced bias eg, if education was used from the death certificate in the numerator and from the population census in the denominator there is the risk that numbers are from somewhat different populations. This may have for example overestimated education inequalities for some Eastern European countries as found in Lithuania [35]. There were differences the degree of missing education data, which may lead to different levels of underestimating inequalities. The older age range of obesity data is likely to have overestimated New Zealand's obesity rates compared to other countries. Small differences in disease definitions and age span were expected to have small effects.

Random error appeared to have contributed to instability in the inequality trends (See supplementary results files for confidence intervals). To limit instability, for some countries risk factor results were averaged across adjacent time periods. Confidence intervals in the Supplementary Files show that in some countries there was reasonably wide uncertainty in the RIIs and SIIs at each time point. However we were interested in the statistical significance of inequality trends. These can be tested with trend statistics, which work for linear (in the chosen outcome scale (e.g. log)) trends, but not otherwise without extension. The compass plots, however, provide a way to see inequality trends. Future developments could include methods to convey statistical certainty (e.g. 90% uncertainty zones around the arrows, or perhaps even a 95% confidence interval about the degrees of travel).

With the exception of smoking-related mortality, it was impossible to directly compare cause-specific mortality trends with those of the related factors given the variety of underlying risk factors contributing to mortality type. This paper therefore focusses on trends and future predictions.

Education is not the only cause of these patterns of social inequality. Children from different socioeconomic positions have very different chances of obtaining a tertiary education, and this might be related to parental education, financial opportunities, racism and other factors. Thus education is likely also a proxy for these factors as well.

Although the RII and SII are superior to more simple measures of inequalities, the interpretation can be complex, as illustrated by Renard, Devleesschauwer [36] who show that under certain conditions improvements in the education distribution can increase the SII/RII.

### Implications practice policy and research

The compass typology provides a useful visual tool for monitoring simultaneous trends in cause specific mortality rates and inequalities, to visualise progress on social health inequalities. Countries who want to address social inequalities can examine their compass typology to examine their progress relative to other countries, the rate of change of these trends over time, and possible future trends—and perhaps even set targets using the compass typology. Risk factor compass typologies can help predict likely changes in future smoking- and obesity- related mortality and social inequalities.

Breaking the link between education and health or mortality outcomes is a policy imperative. Tobacco control should be meeting the needs of the least educated. A greater focus on preventing obesity in the least educated is also important.

### Conclusions

Increasing relative inequalities in smoking prevalence also are suggestive of further increases in relative smoking-related mortality inequalities. Obesity in men and women increased in all countries with an increase in absolute inequalities as well in several countries. There is a concern that these trends will not only hinder future progress in reducing obesity-related mortality (such as CVD and cancer mortality) but also will impact inequalities in obesity-related mortality, although this was not measured here. The compass typology is a useful tool in monitoring country comparisons of trends and inequalities in risk factor prevalence and cause-specific mortality.

### Supporting information

**S1 File. Further methods.**
(PDF)

**S1 Spreadsheet. Risk factor prevalences (subsequently converted into odds), SIIs and RIIs and confidence intervals.**
(XLSX)

**S2 Spreadsheet. Mortality rates, SIIs and RIIs and confidence intervals.**
(XLSX)

**S1 Table. Countries excluded because mortality data was not available.**
(DOCX)

**S2 Table. Countries excluded because mortality inequality data was not available from both the 1980s and 2000s decades.**
(DOCX)

**S3 Table. ICD coding for the cause-specific mortality groups.** *162–163, 165 (ICD-9) and C33–C34, C39 (ICD-10) in most Europe studies in the 1990s, ~not included for NZ smoking

related mortality, ICD coding based on that in: Mackenbach JP, Kulhanova I, Menvielle G, et al. Trends in inequalities in premature mortality: a study of 3.2 million deaths in 13 European countries. *J Epidemiol Community Health*. 2014;69(3):207–217.
(DOCX)

**S4 Table. Compass trajectories for general mortality cause specific inequality trends between 1980 and 2010, and smoking and obesity inequalities between 1990 and 2010.** North is increasing relative inequalities, south is decreasing relative inequalities, west is decreasing prevalence/mortality, and east is increasing prevalence/mortality. South West corresponds with decreasing absolute and relative inequalities and decreasing rates/prevalence. It is the most preferable trend. All countries were required to have data from the first and last decade of the prescribed time periods. The trajectory is taken to be the average linear trend over the whole time period. Where the trajectory was borderline, both possible directions were recorded in the table. Note the distance of the trajectory is not presented in this table and some changes were large and some were very small.
(DOCX)

**S5 Table. Characteristics of mortality, smoking and obesity data.**
(DOCX)

**S6 Table. Source mortality and risk factor datasets, where available.**
(DOCX)

**S1 Fig. Trends in age-standardised smoking prevalence, all education groups combined, 30–79 year olds 1980–2010.**
(PDF)

**S2 Fig. Trends in age-standardised obesity prevalence, 30–79 year olds 1980–2010.**
(PDF)

**S3 Fig. Age-standardized mortality rates of CVD, cancer, external mortality, smoking-related mortality, in men (left) and women (right) 35–79 years.**
(PDF)

**S1 Data.**
(XLSX)

**S2 Data.**
(XLSX)

## Acknowledgments

Thank you Associated Professor James Stanley for statistical advice calculating weighted age-standardised prevalence from New Zealand health survey data. Thank you co-authors and Katalin Kovács, Gwenn Menvielle, Bjorn Heine Strand, Chris White, Johannes Klotz; who supplied the European data.

## Author Contributions

**Conceptualization:** Tony Blakely, Johan P. Mackenbach.

**Data curation:** Andrea Teng, June Atkinson, Ramunė Kalėdienė, Mall Leinsalu, Pekka T. Martikainen, Jitka Rychtaříková.

**Formal analysis:** June Atkinson.

**Funding acquisition:** Andrea Teng, Johan P. Mackenbach.

**Methodology:** Andrea Teng, Tony Blakely.

**Supervision:** Tony Blakely, Johan P. Mackenbach.

**Visualization:** Andrea Teng.

**Writing – original draft:** Andrea Teng.

**Writing – review & editing:** Andrea Teng, Tony Blakely, June Atkinson, Ramunė Kalėdienė, Mall Leinsalu, Pekka T. Martikainen, Jitka Rychtaříková, Johan P. Mackenbach.

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
