## [Decision Letter · Decision Letter 0]

6 Mar 2020

PONE-D-19-33671

Changing social inequalities in smoking, obesity and cause-specific mortality: cross-national comparisons using compass typology

PLOS ONE

Dear Dr. Teng,

Thank you for submitting your manuscript to PLOS ONE. After careful consideration, we feel that it has merit but does not fully meet PLOS ONE’s publication criteria as it currently stands. Therefore, we invite you to submit a revised version of the manuscript that addresses the points raised during the review process.

We would appreciate receiving your revised manuscript by Apr 20 2020 11:59PM. To enhance the reproducibility of your results, we recommend that if applicable you deposit your laboratory protocols in protocols.io, where a protocol can be assigned its own identifier (DOI) such that it can be cited independently in the future. For instructions see: http://journals.plos.org/plosone/s/submission-guidelines#loc-laboratory-protocols

A **rebuttal letter** that responds to **EACH** point raised by the academic editor and reviewer(s). This letter should be uploaded as separate file and labeled 'Response to Reviewers'.A **marked-up copy** of your manuscript that highlights changes made to the original version. This file should be uploaded as separate file and labeled 'Revised Manuscript with Track Changes'.An **unmarked version** of your revised paper without tracked changes. This file should be uploaded as separate file and labeled 'Manuscript'.

We look forward to receiving your revised manuscript.

Kind regards,

Brecht Devleesschauwer

Academic Editor

PLOS ONE

Additional Editor Comments (if provided):

In your revision note, please include EACH comment of the reviewers, provide your reply, and when relevant, include the modified/new text (or motivate why you decided not to modify the text). Note that failure to do so may still result in a rejection of the manuscript.

2)  Please include more information on the surveys and database used. For each country, please ensure that a link or a reference  is provided to the relevant survey/ database used to collect data

3) Our internal editors have looked over your manuscript and determined that it is within the scope of our Health Inequities and Disparities Research Call for Papers. This collection of papers is headed by a team of Guest Editors for PLOS ONE: Clare Bambra, Hans Bosma, Diana Burgess, Joseph Telfair, Barbara Turner, and Jennie Popay. The Collection will encompass a diverse range of research articles on health inequities and disparities.  Additional information can be found on our announcement page: hhttps://collections.plos.org/s/health-inequities

If you would like your manuscript to be considered for this collection, please let us know in your cover letter and we will ensure that your paper is treated as if you were responding to this call. If you would prefer to remove your manuscript from collection consideration, please specify this in the cover letter.

4) We note that you have indicated that data from this study are available upon request. PLOS only allows data to be available upon request if there are legal or ethical restrictions on sharing data publicly. For information on unacceptable data access restrictions, please see http://journals.plos.org/plosone/s/data-availability#loc-unacceptable-data-access-restrictions.

5) PLOS requires an ORCID iD for the corresponding author in Editorial Manager on papers submitted after December 6th, 2016. Please ensure that you have an ORCID iD and that it is validated in Editorial Manager. To do this, go to ‘Update my Information’ (in the upper left-hand corner of the main menu), and click on the Fetch/Validate link next to the ORCID field. This will take you to the ORCID site and allow you to create a new iD or authenticate a pre-existing iD in Editorial Manager. Please see the following video for instructions on linking an ORCID iD to your Editorial Manager account: https://www.youtube.com/watch?v=_xcclfuvtxQ

6) We noted in your submission details that a portion of your manuscript may have been presented or published elsewhere. Please clarify whether this publication was peer-reviewed and formally published. If this work was previously peer-reviewed and published, in the cover letter please provide the reason that this work does not constitute dual publication and should be included in the current manuscript.

Reviewers' comments:

Reviewer's Responses to Questions

**Comments to the Author**

1. Is the manuscript technically sound, and do the data support the conclusions?

Reviewer #1: Yes

Reviewer #2: Yes

Reviewer #3: Yes

2. Has the statistical analysis been performed appropriately and rigorously? 

Reviewer #1: Yes

Reviewer #2: Yes

Reviewer #3: Yes

3. Have the authors made all data underlying the findings in their manuscript fully available?

Reviewer #1: Yes

Reviewer #2: Yes

Reviewer #3: Yes

4. Is the manuscript presented in an intelligible fashion and written in standard English?

Reviewer #1: Yes

Reviewer #2: Yes

Reviewer #3: Yes

5. Review Comments to the Author

Reviewer #1: Review of the manuscript: “Changing social inequalities in smoking, obesity and cause-specific mortality: cross-national comparisons using compass typology”

This is a remarkable paper, applying a typology previously build by the first author and aiming at integrating at a glance different components of health and inequalities and their evolution. In this paper, the authors use this typology to compare 10 countries qua health inequalities and trends for several health indicators.

Only minor changes are suggested. Results could be further developed and a bit restructured for the readability.

Introduction

L94: Please precise that it is the contribution to absolute inequalities

L200: why did you exclude laryngeal cancers from tobacco-related mortality in New Zealand? Did you receive only aggregated rates, or did you compute them yourself, and if so, why to exclude those cancers?

Methods:

It is not clear to me if you started from already computed values, or from the microdata originating from the surveys ?

L174: if you have the microdata, why not choose for the same age-ranges as for the other countries?

Results

As there are really plenty information, I think the readability would be higher is you started, for each topic, with the description of levels and trends of the health indicator, and afterwards the level and trends of inequalities. In addition, I would recommend to fully separating the genders, for each topic.

L290: “Eastern Europe countries tended…”: also Austria.

L295: not only rates in women, but also Rel and absolute ineq. are highest in NZ. Having high relative inequalities when the rate is also high is not so common and is worthwhile to mention, since it denotes very large absolute inequalities.

L307: for CVD mortality, some countries experiences a decrease of absolute inequalities, worthwhile to mention.

L311: please, comment also the decrease in cancer absolute inequalities in several countries for men, and in France, Austria, England for women, here or in the discussion part. Small inequalities in cancer in women can come from the inverse inequality observed in breast cancer.

Discussion

L335 : in Estonia and Lithuania, in men there is not only a decrease of absolute inequalities, but also in relative inequalities. The RII inversed, becoming lower than 1. Could you further discuss this ? Is this the reality, that obesity is more prevalent in higher educated men in those countries, and if so,why ? or could this be an artefact of the data ? the evolution of inequalities in obesity is different in women (decreases in ES, but increases in Lith).

Inequalities seem particularly high in New Zealand, at least for some heath topics. Could you further comment, and maybe provide hypothesis and clues for policies ?

Strength and limitations: the chosen inequality indicators, although include information available from all SE levels, do not adequately reflect shifts in the distribution of the socioeconomic status. Even if the distribution of the socioeconomic status affects the value of the RII and SII, an improvement of the SE distribution is not always (and even mostly not) translated into an improvement of the indicators. Please see discussion of this here https://bmcpublichealth.biomedcentral.com/articles/10.1186/s12889-019-6980-1

Implications practice policy

L405 : “to visualise progress on social inequalities”. I would rather say: “to visualize progress on social health inequalities” . Indeed, the typology cannot show the social inequalities themselves (that are difference in SES itself, for instance magnitude of income differences in, difference in education length/content, differences in wealth), rather it shows differences in health by SES . Social differences create differences in health by SE status.

Fig 1: title on the left side, I would keep ‘Smoking’ and ‘Obesity’ since you do not display the prevalence.

Fig 2: Why is X axis title called “average all exposure mortality rate” and not “smoking-related mortality rate” ?

Reviewer #2: This article uses a compass typology to show trends in rates (or prevalence) and absolute and relative inequalities. The authors examine smoking and obesity prevalence and smoking-related and cause-specific mortality outcomes, across 10 countries. The compass typology seems to me to be a useful way to summarise trends and inequalities simultaneously and possibly predict future trends. I have some specific comments:-

• Last sentence in the abstract – say why the compass typology is useful. Also in first key message. Could add the extension of work to include risk factors as that is new here

• Can you briefly describe the Preston-Glei-Wilmouth method

• Which European Standard Population was used?

• Can you say why education was used here as proxy for socio-economic status. Was this measure easiest to harmonise across countries?

• Which nationally representative surveys were used in each country?

• ‘Compass plots have previously been used only for rates’ – it would be worth citing ref[14] again here, alongside other references if applicable

• Define COPD abbreviation earlier

• Lines 293 and 421 – morality is written instead of mortality

• Figures (1-3b) are blurred and difficult to read

• Mortality data for most countries are presented until around 2011 at the latest, so that international comparisons can be made. There is some scope however, to update the analysis further by examining ‘future’ trends in mortality (deaths up to 2018?) for at least some of the countries. If so, you would be able to see if trends continue as ‘predicted’

• The results shown are for males and females, approximately aged 35-79. SIIs and RIIs vary by age. How are trends likely to differ when looking at different age ranges or groups?

• What are the suggested approaches for preventing smoking/obesity in the least educated groups?

• In the supplementary material, Figure S3 – the values on the y-axis are missing for some female plots

Reviewer #3: The proposed new kind of application of the compass methodology seems to be very promising also for further application. In this work there are some obvious limitations (accurately described) due to the availability of data across all the considered countries and time range. However, it is worth of publication, thanks to the accurate use of the methods and the obtained results.

Some minor revisions are suggested:

- the authors should better reproduce the formulas, maybe with an appropriate editor, only for reading reasons in the final version of the work;

- the authors should consider if a clearer version of the graphs could improve their understanding for the reader. They are clear for me, but I'm not sure that all the interested readers should make the same consideration.

After these revisions, the article can be published.

6. PLOS authors have the option to publish the peer review history of their article (what does this mean?). If published, this will include your full peer review and any attached files.

Reviewer #1: Yes: Françoise Renard

Reviewer #2: No

Reviewer #3: No

---

## [Author Response · Author response to Decision Letter 0]

8 Apr 2020

Department of Public Health

University of Otago Wellington

Wellington, New Zealand

6 April 2020

Dear Editor / Brecht Devleesschauwer

Re: PONE-D-19-33671 Changing social inequalities in smoking, obesity and cause-specific mortality: cross-national comparisons using compass typology.

Many thanks for your request to submit a revised version of this manuscript to PLOS ONE. Please find our response to each of the Journal requirements and Reviewers’ comments below. 

We would be happy for this paper to be considered for the Health Inequities and Disparities Research Call for Papers or as a general publication. 

None of the datasets we used are publically available. There are ethical and legal restrictions on accessing the datasets, to maintain confidentiality for identifiable patient information. Further information on source datasets is now outlined in Table S5a to S5c which is referenced in the manuscript. 

Similar data to that used in this manuscript have been published in peer reviewed journals elsewhere,1-5 however this work does not constitute dual publication. The novel contribution of this manuscript is the use a novel typology to compare inequality trends across cause specific mortalities and risk factors. The contribution that this paper makes is highlighted by all of the reviewers. Previous publications are clearly referenced in the manuscript, and none compared trends across Europe and New Zealand, or combined cause-specific mortality trends with inequality trends in risk factors smoking and obesity. 

In another small change to the submission, I (AT) will be on maternity leave after the 3 April until 9 November. To cover this period, Prof Tony Blakely has agreed to also act as a corresponding author. 

Kind regards, 

Dr Andrea Teng 

(on behalf of the co-authors)

1 Mackenbach, J. P. et al. Trends in inequalities in premature mortality: a study of 3.2 million deaths in 13 European countries. J Epidemiol Community Health 69, 207-217 (2014).

2 Mackenbach, J. P. et al. Changes in mortality inequalities over two decades: register based study of European countries. BMJ 353, i1732, doi:10.1136/bmj.i1732 (2016).

3 Blakely, T., Disney, G., Atkinson, J., Teng, A. & Mackenbach, J. P. A typology for charting socioeconomic mortality gradients:" Go south-west". Epidemiology 28, 594-603, doi:10.1097/EDE.0000000000000671 (2017).

4 Hu, Y. et al. The Impact of Tobacco Control Policies on Smoking Among Socioeconomic Groups in Nine European Countries, 1990-2007. Nicotine Tob Res 19, 1441-1449, doi:10.1093/ntr/ntw210 (2017).

5 Hoffmann, K. et al. Trends in educational inequalities in obesity in 15 European countries between 1990 and 2010. Int J Behav Nutr Phys Act 14, 63, doi:10.1186/s12966-017-0517-8 (2017).

Journal requirements

Authors’ response: To align with journal requirements: we have updated the title page. Figure names in the manuscript have now been abbreviated to ‘Fig 1’ etc. Equations have been numbered. Supplementary files have been renamed and each table has been submitted separately. 

2) Please include more information on the surveys and database used. For each country, please ensure that a link or a reference is provided to the relevant survey/ database used to collect data

Authors’ response: Table S5a to S5c has been added to provide more information on the surveys and databases that were used. 

3) Our internal editors have looked over your manuscript and determined that it is within the scope of our Health Inequities and Disparities Research Call for Papers. This collection of papers is headed by a team of Guest Editors for PLOS ONE: Clare Bambra, Hans Bosma, Diana Burgess, Joseph Telfair, Barbara Turner, and Jennie Popay. The Collection will encompass a diverse range of research articles on health inequities and disparities. Additional information can be found on our announcement page: hhttps://collections.plos.org/s/health-inequities

If you would like your manuscript to be considered for this collection, please let us know in your cover letter and we will ensure that your paper is treated as if you were responding to this call. If you would prefer to remove your manuscript from collection consideration, please specify this in the cover letter.

Authors’ response: We agree to this manuscript being considered for this collection and this is outlined in the cover letter. 

4) We note that you have indicated that data from this study are available upon request. PLOS only allows data to be available upon request if there are legal or ethical restrictions on sharing data publicly. For information on unacceptable data access restrictions, please see http://journals.plos.org/plosone/s/data-availability#loc-unacceptable-data-access-restrictions.

Authors’ response: None of the datasets we used are publically available due to legal and ethical restrictions. Table S5a to S5c provides further details on the datasets that were used. 

Authors’ response: A large number of datasets are used in this study. There are legal and ethical restrictions on sharing all of these de-identified datasets which differ between countries. Please see Table S5a to S5c for further information on the datasets that were used. 

Authors’ response: NA, as above. 

5) PLOS requires an ORCID iD for the corresponding author in Editorial Manager on papers submitted after December 6th, 2016. Please ensure that you have an ORCID iD and that it is validated in Editorial Manager. To do this, go to ‘Update my Information’ (in the upper left-hand corner of the main menu), and click on the Fetch/Validate link next to the ORCID field. This will take you to the ORCID site and allow you to create a new iD or authenticate a pre-existing iD in Editorial Manager. Please see the following video for instructions on linking an ORCID iD to your Editorial Manager account: https://www.youtube.com/watch?v=_xcclfuvtxQ

Authors’ response: AT has added her ORCID to her PLOS ONE profile.

6) We noted in your submission details that a portion of your manuscript may have been presented or published elsewhere. Please clarify whether this publication was peer-reviewed and formally published. If this work was previously peer-reviewed and published, in the cover letter please provide the reason that this work does not constitute dual publication and should be included in the current manuscript.

Authors’ response: Please see the cover letter. 

Reviewer #1: 

Review of the manuscript: “Changing social inequalities in smoking, obesity and cause-specific mortality: cross-national comparisons using compass typology”

This is a remarkable paper, applying a typology previously build by the first author and aiming at integrating at a glance different components of health and inequalities and their evolution. In this paper, the authors use this typology to compare 10 countries qua health inequalities and trends for several health indicators.

Authors’ response: Many thanks for reviewing this paper and thank you for highlighting the value in the typology approach. 

Only minor changes are suggested. Results could be further developed and a bit restructured for the readability.

Authors’ response: Please see responses below. 

Introduction

L94: Please precise that it is the contribution to absolute inequalities

Authors’ response: ‘absolute’ has been added to the description. 

L200: why did you exclude laryngeal cancers from tobacco-related mortality in New Zealand? Did you receive only aggregated rates, or did you compute them yourself, and if so, why to exclude those cancers?

Authors’ response: This was due to the way laryngeal cancers had been aggregated in the existing census mortality linked dataset. The original mortality ICD codes were not available in the linked datasets because they had been aggregated before data linkage was done, for consistency and confidentiality reasons with small numbers. Therefore, it would have been a slow and complicated process, and possibly not successful due to the length of time since data linkage, to have been able to get the external body, ie, Stats NZ, to put this information on the files for us, with little potential additional gain.

Methods:

It is not clear to me if you started from already computed values, or from the microdata originating from the surveys ?

L174: if you have the microdata, why not choose for the same age-ranges as for the other countries?

Authors’ response: NZ and European smoking and obesity analyses were largely done for the 30-79 year old age group. However some countries did not have data for the very oldest age groups. We have corrected the age ranges in the text for obesity because previously cited age groups as used in the Hoffman paper were incorrect. Sentence now reads; “This was done for survey participants 30-79 years old, with the exception of Hungary and Lithuania where the upper age limit was 64 years old.”

Results

As there are really plenty information, I think the readability would be higher is you started, for each topic, with the description of levels and trends of the health indicator, and afterwards the level and trends of inequalities. In addition, I would recommend to fully separating the genders, for each topic.

Authors’ response: Please see Results for change in order of results description and clearer separation of men and women in descriptions. We believe the comparison between men and women is useful and therefore some of these comparisons have been left in. 

Space limitations prevented a thorough description of levels and trends of every health indicator. 

L290: “Eastern Europe countries tended…”: also Austria.

Authors’ response: Yes the relative inequalities were low in Austria but because there was no evidence they crossed over from RR<1 to RR>1 we have left this sentence as it stands. 

L295: not only rates in women, but also Rel and absolute ineq. are highest in NZ. Having high relative inequalities when the rate is also high is not so common and is worthwhile to mention, since it denotes very large absolute inequalities.

Authors’ response: New sentence reads: “The lowest smoking related mortality rates in women were in Lithuania and the highest were in New Zealand which also experienced the highest absolute and relative inequalities.” 

L307: for CVD mortality, some countries experiences a decrease of absolute inequalities, worthwhile to mention.

Authors’ response: Thanks, this sentence has now been added; “There was a decrease in absolute inequalities in men and women from England and Wales, Finland, and France.”

L311: please, comment also the decrease in cancer absolute inequalities in several countries for men, and in France, Austria, England for women, here or in the discussion part. Small inequalities in cancer in women can come from the inverse inequality observed in breast cancer.

Authors’ response: Thank you, a new sentence has been added to the results: “There was a decrease in absolute inequalities in cancer for men and women in England and Wales, and for men in France.” 

On our reading of the typologies there was no decrease in absolute inequalities (crossing of the contours) in France, Austria, England for women. We also note that the pattern of inequalities in breast cancer has reversed in some countries, to become disproportionately greater in low SEP/education groups, possibly affecting the differences between men and women. 

Discussion

L335 : in Estonia and Lithuania, in men there is not only a decrease of absolute inequalities, but also in relative inequalities. The RII inversed, becoming lower than 1. Could you further discuss this ? Is this the reality, that obesity is more prevalent in higher educated men in those countries, and if so,why ? or could this be an artefact of the data ? the evolution of inequalities in obesity is different in women (decreases in ES, but increases in Lith).

Authors’ response: We have added some further clarification to the Discussion, also referencing a previous study and hypotheses for these results. The sentence in the Discussion was revised to read: “This was combined with increasing absolute inequalities in many countries (eastward trend) with two countries reporting decreased a decrease in absolute and relative inequalities where obesity became more prevalent in men with higher education (Lithuania and Estonia) but this pattern was not noted for woman. This finding has been seen previously in Estonia [29] and the authors’ hypothesise physical activity from manual occupations as a contributing factor. It also may be related to being on an earlier stage in the nutrition transition [30], where energy dense food became more accessible to the most educated first after the dissolution of the Soviet Union, and then later became more concentrated in the least educated.” 

Inequalities seem particularly high in New Zealand, at least for some health topics. Could you further comment, and maybe provide hypothesis and clues for policies ?

Authors’ response: NZ did have high absolute and relative inequalities in obesity prevalence, cancer in women and smoking-related mortality in women. NZ is known to historically have had high rates of smoking in women and high levels of inequality in this group (high Māori women smoking rates). These patterns may have contributed to the cancer and smoking-related mortality inequalities in NZ women. The impact of historical trends in smoking is discussed in the text. Also the worsening inequalities in obesity in NZ (and two other countries) are discussed from Line 338, 2nd paragraph in the discussion, which have also been linked to cancer inequalities (described in text). 

Strength and limitations: the chosen inequality indicators, although include information available from all SE levels, do not adequately reflect shifts in the distribution of the socioeconomic status. Even if the distribution of the socioeconomic status affects the value of the RII and SII, an improvement of the SE distribution is not always (and even mostly not) translated into an improvement of the indicators. Please see discussion of this here https://bmcpublichealth.biomedcentral.com/articles/10.1186/s12889-019-6980-1

Authors’ response: The interpretation of (changes in) the RII and SII is indeed complex. The Renard et al. paper focusses on a problem that is not directly relevant to our analysis: they warn against using the RII/SII when "improving the educational attainment is part of a policy", and highlight that under certain conditions an increase in the percentage of high educated will push the RII/SII up. We have however added this sentence to the limitations:

"Although the RII and SII are superior to more simple measures of inequalities, the interpretation can be complex, as illustrated by Renard, Devleesschauwer (35) who show that under certain conditions improvements in the education distribution can increase the SII/RII.

Implications practice policy

L405 : “to visualise progress on social inequalities”. I would rather say: “to visualize progress on social health inequalities” . Indeed, the typology cannot show the social inequalities themselves (that are difference in SES itself, for instance magnitude of income differences in, difference in education length/content, differences in wealth), rather it shows differences in health by SES. Social differences create differences in health by SE status.

Authors’ response: Thank you, we agree and this change has been made.

Fig 1: title on the left side, I would keep ‘Smoking’ and ‘Obesity’ since you do not display the prevalence.

Authors’ response: We agree, this change has been made. 

Fig 2: Why is X axis title called “average all exposure mortality rate” and not “smoking-related mortality rate” ?

Authors’ response: The mortality rate is the average across the three levels of education exposure. 

Reviewer #2: 

This article uses a compass typology to show trends in rates (or prevalence) and absolute and relative inequalities. The authors examine smoking and obesity prevalence and smoking-related and cause-specific mortality outcomes, across 10 countries. The compass typology seems to me to be a useful way to summarise trends and inequalities simultaneously and possibly predict future trends. I have some specific comments:-

Authors’ response: Many thanks for reviewing this paper. 

• Last sentence in the abstract – say why the compass typology is useful. Also in first key message. Could add the extension of work to include risk factors as that is new here

Authors’ response: Thank you. Abstract and key message have been changed to read “The compass typology was useful to compare trends in inequalities because it simultaneously tracks changes in rates/odds, and absolute and relative inequality measures.” Additional sentence added to this key message “The typology was extended for the first time to include risk factors as well as cause-specific mortality.”

• Can you briefly describe the Preston-Glei-Wilmouth method

Authors’ response: This sentence in the Introduction was updated and now reads “For example the Preston-Glei-Wilmoth method relies on lung cancer mortality as an indicator of smoking prevalence and was used to estimate the contribution of smoking to absolute educational inequalities in all-cause mortality in 14 European countries from 1990-2004” 

• Which European Standard Population was used?

Authors’ response: 1976. This is now indicated in the Method’s description of standardisation.

• Can you say why education was used here as proxy for socio-economic status. Was this measure easiest to harmonise across countries?

Authors’ response: This sentence was added to the methods: “Education was selected as a proxy for socioeconomic position given its widespread availability; ease of harmonization across countries and lower sensitivity to health-related social mobility than other socioeconomic measures.” 

• Which nationally representative surveys were used in each country?

Authors’ response: Tables S5a to S5c hasve been added to the supplementary material to outline the datasets and surveys used in each country and this is now referenced in the methods; “For more information on the source mortality 1-3 and survey risk factor 4,5 datasets please see previous publications and Supplementary Tables S5a to S5c.” 

• ‘Compass plots have previously been used only for rates’ – it would be worth citing ref[14] again here, alongside other references if applicable

Authors’ response: Ref 14 has now been cited. 

• Define COPD abbreviation earlier

Authors’ response: Thanks, this has now been done. 

• Lines 293 and 421 – morality is written instead of mortality

Authors’ response: This has now been corrected. Thanks. 

• Figures (1-3b) are blurred and difficult to read

Authors’ response: Apologies, improved versions of these will be updated on the revised submission . 

• Mortality data for most countries are presented until around 2011 at the latest, so that international comparisons can be made. There is some scope however, to update the analysis further by examining ‘future’ trends in mortality (deaths up to 2018?) for at least some of the countries. If so, you would be able to see if trends continue as ‘predicted’

Authors’ response: This could be the focus of a second paper, perhaps when the next decade of data is available for these types of analysis. Delays between death and data availability can prevent earlier analyses. 

• The results shown are for males and females, approximately aged 35-79. SIIs and RIIs vary by age. How are trends likely to differ when looking at different age ranges or groups?

Authors’ response: Age differences are something that was examined in a previous typology paper (Blakely, T., Disney, G., Atkinson, J., Teng, A. & Mackenbach, J. P. A typology for charting socioeconomic mortality gradients:" Go south-west". Epidemiology 28, 594-603 (2017).) Mortality trends were likely to occur at higher rates in older age groups/ranges, and this point is acknowledged in our limitations. Inequality trends in all-cause mortality in the above paper however were similar across age groups, so small age group variations may have a limited effect on the observed inequality trends. 

• What are the suggested approaches for preventing smoking/obesity in the least educated groups?

Authors’ response: Ideally policy implications should follow from the data and analyses we have carried out. We have not evaluated smoking or obesity prevention policies here, so we have not made any recommendations about these. Also, further research is also needed to understand how nutrition policies such as fiscal measures, addressing availability and marketing restrictions may affect the least educated groups and health inequalities. 

• In the supplementary material, Figure S3 – the values on the y-axis are missing for some female plots

Authors’ response: Thank you, this has now been corrected. 

Reviewer #3: 

The proposed new kind of application of the compass methodology seems to be very promising also for further application. In this work there are some obvious limitations (accurately described) due to the availability of data across all the considered countries and time range. However, it is worth of publication, thanks to the accurate use of the methods and the obtained results.

Authors’ response: Many thanks for reviewing this paper. 

Some minor revisions are suggested:

- the authors should better reproduce the formulas, maybe with an appropriate editor, only for reading reasons in the final version of the work;

Authors’ response: Current equations are produced in word equation editor and font can be increased as necessary. Please let us know if there is a preferred format for submitting equations and we would be happy to revise. 

- the authors should consider if a clearer version of the graphs could improve their understanding for the reader. They are clear for me, but I'm not sure that all the interested readers should make the same consideration.

Authors’ response: Please see response to reviewer above. Quality of figures has now been improved via the submission process. 

After these revisions, the article can be published.

---

## [Editor Report · Decision Letter 1]

27 Apr 2020

Changing social inequalities in smoking, obesity and cause-specific mortality: cross-national comparisons using compass typology

PONE-D-19-33671R1

Dear Dr. Teng,

We are pleased to inform you that your manuscript has been judged scientifically suitable for publication and will be formally accepted for publication once it complies with all outstanding technical requirements.

With kind regards,

Brecht Devleesschauwer

Academic Editor

PLOS ONE

Additional Editor Comments (optional):

Thank you for addressing the reviewer comments.
---

## [Editor Report · Acceptance letter]

25 Jun 2020

PONE-D-19-33671R1 

Changing social inequalities in smoking, obesity and cause-specific mortality: cross-national comparisons using compass typology 

Dear Dr. Teng:

I'm pleased to inform you that your manuscript has been deemed suitable for publication in PLOS ONE. Congratulations! Your manuscript is now with our production department. 

Kind regards, 

on behalf of

Prof. Dr. Brecht Devleesschauwer 

Academic Editor

PLOS ONE